# Effect of High-Intensity Interval Training on Quality of Life, Sleep Quality, Exercise Motivation and Enjoyment in Sedentary People with Type 1 Diabetes Mellitus

**DOI:** 10.3390/ijerph182312612

**Published:** 2021-11-30

**Authors:** Jesús Alarcón-Gómez, Iván Chulvi-Medrano, Fernando Martin-Rivera, Joaquín Calatayud

**Affiliations:** 1Department of Physical Education and Sports, University of Valencia, 46010 Valencia, Spain; jesusadol18@gmail.com; 2UIRFIDE Sport Performance and Physical Fitness Research Group, University of Valencia, 46010 Valencia, Spain; ivan.chulvi@uv.es; 3Research Group in Prevention and Health in Exercise and Sport, University of Valencia, 46010 Valencia, Spain; 4Exercise Intervention for Health Research Group (EXINH-RG), Department of Physiotherapy, University of Valencia, 46010 Valencia, Spain; joaquin.calatayud@uv.es

**Keywords:** diabetes type 1, HIIT, sleep quality, exercise motivation, quality of life

## Abstract

(1) Background: Type 1 diabetes mellitus (T1DM) people’s health-related quality of life (HRQoL) is affected by glycemic control. Regular exercise is strongly recommended to these patients due to its cardiovascular and metabolic benefits. However, a large percentage of patients with T1DM people present a sedentary behavior because of the fear of a post-exercise hypoglycemia event, lack of time, lack of motivation and the complicated management of exercise, glycemic and insulin dose interaction. High-intensity interval training (HIIT) is an efficient and safe methodology since it prevents hypoglycemia and does not require much time, which are the main barriers for this population to doing exercise and increasing physical conditioning. (2) Methods: Nineteen sedentary adults (37 ± 6.5 years) with T1DM, were randomly assigned to 6 weeks of either HIIT (12-16-20 × 30-s intervals interspersed with 1-min rest periods) performed thrice weekly, or to the control group, which did not train. HRQoL, sleep quality, exercise motivation and enjoyment were measured as psychological variables. (4) Results: HRQoL improved in physical and social domains, PF (1.9%); PR (80.3%); GH (16.6); SF (34.1%). Sleep quality improved in the HIIT group by 21.4%. Enjoyment improved by 7% and intrinsic motivation was increased by 13%. (5) Conclusions: We suggest that the 6-week HIIT program used in the present study is safe, since no severe hypoglycemia were reported, and an effective strategy in improving HRQoL, sleep quality, exercise motivation and enjoyment which are important psychological well-being factors in T1DM people.

## 1. Introduction

Type 1 diabetes mellitus (T1DM) is a chronic metabolic disorder characterized by the insufficient production of endogenous insulin due to pancreatic beta cells autoimmune destruction which is associated with multiple clinical manifestations [1]. Reports from the International Diabetes Federation and the World Health Organization suggest that 25–45 million adults (>20 years old) suffer from T1DM worldwide [2]. Furthermore, it is estimated that the number of people with T1DM will increase by 25% by 2030 [2,3].

T1DM generates several negative consequences affecting health-related quality of life (HRQoL) [4]. For instance, frequent self-monitoring of blood glucose, worry about hypoglycemia and lifestyle changes management derived from the disease increase stress situations, depression, anxiety and fear [5,6]. In addition, sleep quality (a determinant factor in glycemic control and physical and mental well-being) is decreased among T1DM patients, with shorter sleep duration and more episodes of apnea than healthy people [7].

Treatment of T1DM requires a rigorous balance among diet, physical activity and exogenous insulin administration to maintain blood glucose in normal ranges [8]. From all these factors, exercise is commonly known as an essential tool to improve HRQoL among those with T1DM [9]. However, exercise has shown some negative effects on sleep quality in this population due to the increase of exercise-induced nocturnal hypoglycemia [6]. T1DM people are mostly sedentary due to, mainly, the following reasons: (a) fear of hypoglycemia, (b) lack of time and (c) lack of motivation [10,11,12]. Given this background, new exercise methods eliciting enjoyment and motivation are needed to increase adherence, and consequently, improve HRQoL of T1DM people, helping to avoid sleep disturbances. Thus, the aforementioned barriers that T1DM people face might be overcome with high-intensity interval training (HIIT). This training method involves repeated brief bouts of high intensity (>85% VO_2_max) intermitted with passive or active recovery periods, requiring lower exercise duration than moderate-intensity continuous training (MICT) [13,14]. In addition, HIIT also prevents hypoglycemic events, typical of MICT, due to its anaerobic predominance, avoiding nocturnal hypoglycemia as well [12]. There is also evidence to suggest that HIIT elicits at least the same psychological effects, including enjoyment, as MICT among healthy and pathologic populations such as obesity, Type 2 diabetes and cardiovascular disease [15,16,17,18]. However, there are no studies investigating its possible application to improve psychological well-being and exercise adherence in T1DM people. The safety and time efficiency of HIIT make this exercise method an interesting alternative for this population. So far, HIIT was only applied in T1DM people to analyze the long-term effects in aerobic capacity and glycemic control [19,20,21]. Therefore, the aim of this study was to analyze the effect of HIIT on variables that influence psychological well-being in T1DM individuals, previously inactive.

## 2. Materials and Methods

### 2.1. Participants and Research Design

Nineteen sedentary adults clinically diagnosed as T1DM (10 males and 9 females) from the Valencian Diabetes Association (VDA) and social media announcement were selected to be part of the sample (Table 1). The following inclusion criteria were set: (1) aged 18–45 years, (2) duration of T1DM > 4 years, (3) HbA1C < 10% (4) no structured exercise training programs in the previous 6 months, (5) no known comorbidities related or not to diabetes. Smoking regularly, taking any medication that affects heart rate, suffering overweight or obesity and having planned major surgery were adopted as exclusion criteria (Figure 1). The institutional gym was the lab where all the activities were conducted. Participants were informed of the purposes and risks involved in the study before giving their informed written consent to participate. Furthermore, they completed two questionnaires before the beginning of the measurement protocols: the PAR-Q to assess participants’ level of risk to safely participate and the IPAQ (short version), to ensure the previous sedentary behavior of the subjects. All the procedures were developed in accordance with the principles of the Declaration of Helsinki and were approved by the local Institutional Review Board (H1421157445503).

This is a randomized controlled trial with parallel design in which the eligible subjects were randomly allocated by the researchers (www.randomizer.org, accessed on 2 January 2019) to the experimental or to the control group, and classified by gender to ensure a balanced number of men and women in each group. They all were asked to maintain their habitual diet and not to exercise outside of the study.

### 2.2. Measures

All tests were completed online before starting the experimental period and after the last training session with a previous period of instruction to the participants to ensure the total comprehension of the questionnaires which were used. Nonetheless, patients were instructed to ask any doubt that might arise to the investigators while the questionnaires were being completed. All the procedures were exactly conducted in the same way both times.

#### 2.2.1. Health-Related Quality of Life

Health-related quality of life (HRQoL) was self-reported by completing the short form 36 health survey (SF-36) [22]. This questionnaire is a valid and reliable generic instrument to assess HRQoL and its components [23,24]. This questionnaire contains 36 questions including an eight-domain profile of functional health and well-being scores (physical functioning, role limitation due to physical problems, bodily pain, general health, vitality, social functioning, role limitation due to emotional problems and mental health). For each parameter, scores are coded, summed and transformed to a scale from 0 (the worst possible condition) to 100 (the best possible condition) [25,26].

#### 2.2.2. Sleep Quality

Sleep quality was evaluated by the Pittsburgh Sleep Quality Index (PSQI), a clinical sleep-behavior questionnaire that has been validated for use in patients with different chronic diseases and the general population [27,28]. The PSQI is an instrument that aims to measure subjective sleep quality and related disorders and involves seven domains: (i) subjective sleep quality (very good to very bad), (ii) sleep latency (≤15 min to >60 min), (iii) sleep duration (≥7 h to <5 h), (iv) sleep efficiency (≥85% to <65% h sleep/h in bed), (v) sleep disturbances (not during the past month to ≥3 times per week), (vi) use of sleeping medications (none to ≥3 times a week), and (vii) daytime dysfunction (not a problem to a very big problem), divided into 10 questions, of which questions 1–4 are open and 5–10 are semi-open [29]. The PSQI scoring scale ranges from 0 to 21. Each domain has a set weight between zero and three and the global score is given by the sum of the scores in the seven domains [30]. A PSQI global score higher than 5 indicates poor sleep quality [31].

#### 2.2.3. Exercise Motivation

The behavioral regulation in the exercise questionnaire (BREQ-2) was used to measure participants’ underlying motivational regulation relating to HIIT participation [32]. It is comprised of five subscales: (1) intrinsic (e.g., “I exercise because it’s fun”); (2) identified (e.g., “I value the benefits of exercise”); (3) introjected (e.g., “I feel guilty when I don’t exercise”); (4) external (e.g., “I exercise because other people say I should”); and (5) amotivation (e.g., “I don’t see why I should have to exercise”) [32,33,34]. A 5-point Likert scale ranging from; 1 = “not true for me” to 5 = “very true for me” is used to rate each of its 19 items with the generation of each subscale score based on mean score across subscale items [35].

#### 2.2.4. Exercise Enjoyment

To assess HIIT enjoyment, participants were asked to complete the 16-item Physical Activity Enjoyment Scale (PACES) [36]. This instrument consists of questions relating to enjoyment of exercise with the instruction “When I am active…”. This 16-item survey is scored on a 5-point bipolar scale from 1 (totally disagree) to 5 (totally agree). Example items include “I enjoy it/I hate it”, “I find it energizing/I find it tiring”, “It gives me a strong sense of accomplishment/It does not give me any sense of accomplishment at all” [37]. The score was obtained with the sum of the positive elements and the restoration of the negative elements with higher scores indicating higher levels of enjoyment [36].

### 2.3. Exercise Interventions

Initially, an incremental test on a cycle ergometer was performed by all the participants (Excite Unity 3.0, Technogym S.p.A, Cesena, Italia) to determine peak power output (PPO) and peak oxygen consumption (VO_2_peak) using a gas collection system (PNOE, Athens, Greece) that was calibrated in each test by means of ambient air [38]. Capillary blood glucose concentrations were checked before the commencement of the incremental test by their own blood glucose monitoring devices. They were told to arrive at the lab with a glycemic level >100 mg/dL and less than 300 mg/dL in absence of ketones. If the glycemic level was optimum, the participant began the test normally. If not, the intake of 15–30 g of fast-acting carbohydrates (CHO) we had available was compulsory when glycemia was <100 mg/dL and a small corrective insulin dose was used if hyperglycemia appeared without ketones. In the presence of ketones, the exercise was canceled. Exercise was not allowed to start until blood glucose was correct. In the same way, it was recommended that patients not exercise at the peak of insulin action [39].

Firstly, a warm-up of 5 min at 40 Watts (W) was performed. After that, the workload was increased by 20 W every minute until volitional exhaustion. Participants were verbally encouraged to give their maximum effort during the test. The test ends with a cool down of 5 min at 40 W. Monitoring of heart rate was carried out by a Polar H10 (Polar Electro, Kempele, Finland). VO_2_peak was taken as the highest mean achieved within the last 15 s prior to exhaustion. Peak power output was registered to individualize the workloads in the HIIT protocol.

The hour of the day that each subject completed the test was recorded, as well as the menstrual phase of each female participant with the aim of repeating the same conditions in the second measurement to prevent their interference on the outcomes.

The training period was initiated the week after the pre-experimental measurements. Participants of the experimental group trained thrice weekly for 6 weeks under researcher supervision on a cycle ergometer (Excite Unity 3.0, Technogym S.p.A, Cesena, Italy). Heart rate while exercising was monitored with a Polar H10 (Polar Electro, Kempele, Finland) that was preconfigured with their heart rate zones. The HIIT protocol performed was a type 1:2, which means that the high-intensity intervals lasted exactly half the time that the rest intervals did. The saddle height was always adjusted to the height of the subject’s iliac crest. The training session began with a 5-min warm-up at 50 W. Then, they performed 30-s sets of high-intensity cycling at a workload selected to elicit 85% of their individual PPO interspersed with 1 min of active recovery at 40% PPO (Figure 2). The number of high-intensity intervals increased from twelve reps in weeks 1 and 2, to sixteen reps in weeks 3 and 4, to twenty reps in weeks 5 and 6. Training ends with a 5-min cool down performed at 50 W. After the session, participants were told to check their glycemia level frequently (every 1–2 h) and notify the investigators if a glycemia drop below 70 mg/dL occurred during the 24 h following the exercise even though felt well.

All sessions were supervised by the investigators and participants were asked not to fast before the training and not to exercise within 30 min of a meal, aiming to perform the HIIT under real-world conditions. For that reason, no adjustments were made in insulin dose. Glucose levels were checked at least before and immediately after each exercise session, it was re-checked when glucose was not in the safe range (100–250 mg/dL). Fast-acting carbohydrates (15–30 g) were ingested when glycemia fell to ≤100 mg/dL. Hyperglycemia (250–300 mg/dL) was not set as a reason for postponing exercise if the patient felt well and ketones were negative [19,21]. Subjects assigned to the control group were instructed to maintain their current lifestyle and dietary intakes during the study period.

### 2.4. Statistical Analysis

All variables were expressed as a mean and standard deviation and were analyzed using a statistical package (SPSS Inc., Chicago, IL, USA). Normality assumption by Shapiro-Wilks was checked for each variable. A mixed factorial ANOVA (2 × 2) was performed to assess the influence of “condition” (i.e., control group vs. experimental group) and “time moment” variable (i.e., pre-intervention, post-intervention). In the event that the Sphericity assumption was not met, freedom degrees were corrected using Greenhouse–Geisser estimation. Post-hoc analysis was corrected using Bonferroni adjustment. The Wilcoxon and Mann–Whitney U tests were used for the variables that did not meet the normality (quality of life and exercise motivation). Cohen’s D was used to assess the magnitude of mean differences between control vs. experimental conditions.

## 3. Results

### 3.1. Health-Related Quality of Life

The Mann–Whitney U and Wilcoxon tests showed significant differences between pre–post measurements of the following variables in the experimental group: Physical functioning, Physical role limitations, Pain, General health, Energy/Fatigue, Social functioning. In the control group, there were no differences between pre–post measurements. See Table 2.

### 3.2. Sleep Quality

ANOVA revealed significant statistical differences for the main effect of “pre–post” (F_(1,17)_ = 28.03; *p* < 0.001), pair comparations showed significant differences between pre–post measurements in the experimental group. In the control group, there were no differences between pre–post measurements. See Table 3.

### 3.3. Exercise Motivation

The Mann–Whitney U and Wilcoxon tests showed significant differences between pre–post measurements of the following variables in the experimental group: Intrinsic, identified, Introjected, External, Amotivation. In the control group, there were no differences between pre–post measurements. See Table 4.

### 3.4. Exercise Enjoyment

ANOVA revealed significant statistical differences for the main effect of “pre–post” (F_(1,17)_ = 21.92; *p* < 0.001), pair comparations showed significant differences between pre–post measurements in the experimental group. In the control group, there were no differences between pre–post measurements. See Table 5.

### 3.5. Adverse Events

There were three mild hypoglycemia cases (67.9 ± 2.6 mg/dL) of 198 total trainings (1.5%), occurring immediately after exercise which only required a few minutes of rest and carbohydrate ingestion to be solved. No adverse cardiac events, respiratory events or musculoskeletal injuries were reported in the experimental period. There were no episodes of hyperglycemia, nocturnal hypoglycemia or episodes of diabetic ketoacidosis.

## 4. Discussion

The main result of the study shows that a 6-week HIIT is sufficient to improve well-being and exercise adherence in the previously inactive T1DM population, since HRQoL, SQ, enjoyment and exercise motivation obtained better results in the experimental group. Moreover, the study showed that this training method is safe for this population since no insulin or carbohydrate intake adjustments were made. Moreover, only 3 of 198 total trainings, which means less than 1.5%, resulted in hypoglycemia, and they were mild cases (69.7 ± 2.6 mg/dL). No severe hyperglycemias were reported. These data suggest that HIIT prevents hypoglycemia as well as previous studies reported [12,21].

Our findings show that the HIIT group experimented gave significant HRQoL improvements in the physical and social domains, with better results than the control group at PR, BP, VT and SF domains. Previous studies among T1DM people—albeit mainly focused on children, adolescents and young adults—mostly reported better HRQoL when physical exercise is part of their lifestyle or after a training period [40,41], although it can also be perceived as a stressor and have no positive effects on this outcome [42]. Previous reviews which analyzed the effect of HIIT on HRQoL among heart failure patients—one of the most recurrent T1DM complications [43]—found that all dimensions improved after a period of HIIT. There was high heterogeneity in HIIT protocols, but it is worth mentioning that investigations that reported mental improvements in HRQoL used protocols of at least 10 weeks of duration [44,45]. In the present study, a 6-week HIIT was conducted, and thus, the lower exercise volume in our study could explain the absence of improvement in the mental components of HRQoL. Nevertheless, our results are in line with those obtained by Stavrinou and colleagues [46], who showed improvements in the physical components of the SF-36 but not in the mental dimension after a similar HIIT protocol conducted during 8 weeks among inactive healthy people.

To the best of our knowledge, there is only one previous publication that analyses HRQoL in T1DM after a HIIT protocol [47]. However, the aforementioned study used obese or overweight participants and the training consisted of 4 bouts of 4 min at 85–95% HR_peak_ interspersed with 3-min recovery intervals at 50–70% HR_peak_ for 12 weeks. The results from this study suggest no changes in HRQoL in the studied population, in contrast with our results. However, the previously mentioned study used a different questionnaire (Diabetes Quality of Life questionnaire), which might explain the different results. We used the SF-36 because analyzes domains of physical functionality and psychological well-being, while the Diabetes Quality of Life questionnaire predicts care behaviors and satisfaction with diabetes control [48] which is less specific/related to physical exercise domains.

Relevantly, we found that the HIIT group had a greater PSQI global score than the control group, with a 21.3% increase, which means moving from previous “poor sleep quality” to “good sleep quality” [49].

To date, the previous literature has focused on the effect of exercise on nocturnal hypoglycemia due to its relevance on sleep quality in T1DM, not directly measuring sleep quality measurement. Given that aerobic exercise has been shown to increase the likelihood of nocturnal hypoglycemia in T1DM people [50], recent studies have shown HIIT as an interesting training strategy to prevent nocturnal hypoglycemia, especially, if performed early in the morning [12,50]. This could explain the sleep quality improvement shown in our investigation of the HIIT group, insomuch as no severe hypoglycemia were reported, including nocturnal periods. Sleep disorders have a negative influence on mental and physical health and decrease the quality of life in T1DM. HIIT could be postulated as an interesting strategy to improve sleep quality, in part due to the reduction of post-exercise and nocturnal hypoglycemia preventing night-time awakenings and consequently, positively affect HRQoL in this population.

In reference to exercise type HIIT enjoyment, our results showed a greater improvement (4.1%) in the HIIT group. This result is in line with previous studies which reported that HIIT enhances exercise enjoyment in both inactive but healthy and people with pathologies such as obesity or Crohn’s disease, using in most of them the PACES test to determine exercise enjoyment [17,51,52].

Lack of motivation is presented as one of the most important barriers of T1DM people to exercising [53]. We found that the used HIIT protocol increased self-determined domains of autonomous motivational regulation: intrinsic (11.5%) and identified (7.6%) dimensions. Conversely, external motivation (34.7%) and demotivation (46.6%) components provided lower scores (*p* < 0.05), which support the self-determined improvement. In agreement with these results, a previous study [54] found that autonomous regulation towards exercise was improved after only one HIIT session. Furthermore, a 10-month HIIT protocol including core and functional exercises performed by inactive obese women reported increased intrinsic regulation (33%) and identified regulation (88%) and decreased external regulation by 75%. These data are congruent with the obtained in the present study but with a higher effect, probably due to the greater duration of the intervention. Reasons that may explain that HIIT increases exercise motivation are related to different exercise effects and the special characteristics of HIIT.

The self-determination theory [55] considers three basic psychological needs (autonomy, competency and relatedness). Their satisfaction influences exercise motivation. In this way, this theory indicates that people show greater intrinsic regulation (“I do exercise because I enjoy it”) if they perceived some decision capacity, efficacy in the task they are performing and good social relationship with people around them. Nevertheless, if the mentioned psychological needs are frustrated, probably the individual shows an external regulation (“I do exercise to get something, not because I like it”) or demotivation [56]. We hypothesize that the protocol increased the most self-determined regulations and reduced the most external dimensions due to three main reasons: (i) Given that the protocol was a 1:2 HIIT, participants could complete the sessions effectively but with a high level of exigency, increasing their competency subjective perception; (ii) They were free to choose the hour of the training, listen to music or not and they had no strict instructions related to insulin or carbohydrate intake. Those factors could influence their autonomy perception; (iii) Participants were in touch between them, physically and through social media. They talked about the investigation and arranged meetings outside the lab. Those aspects could positively affect their social relationship [18,55]. Given that enjoyment and self-determined domains of motivation are highly correlated with adherence [17,57,58], it could be proposed that HIIT elicits exercise adherence and consequently, contribute physiological and psychological benefits to this population.

Since there are no previous studies with similar characteristics our results should be taken with caution. No medical specialists were included in the research group which would have contributed to a better quality in the metabolic adjustments. However, the medical services of the university were aware of the investigation development. Although the statistical power analysis performed with G*power 3.0 resulted in values higher than 0.80 for all variables and tests performed, the small sample size remains the main limitation. Furthermore, we did not use specific HRQoL questionnaires for T1DM or objective sleep quality measurements. However, the used questionnaires are valid and reliable, providing novel data in this population. The nutritional habits of the participants were not measured despite being asked to maintain them with no change. Future longer interventional studies which also compare additional interventions are needed to corroborate our results and add further insight into the effects of using HIIT among T1DM people.

## 5. Conclusions

In conclusion, a 6-week HIIT protocol 1:2 type, performing high-intensity intervals at 85% PPO and active rest intervals at 40% PPO in a cycle ergometer during three sessions per week, apart from being accessible and safe since participants were able to complete all the sessions with the intensity required without suffering any severe undesirable episode of hypoglycemia, was sufficient to improve HRQoL, sleep quality, exercise enjoyment and exercise motivation in previously inactive T1DM people.

## Figures and Tables

**Figure 1 ijerph-18-12612-f001:**
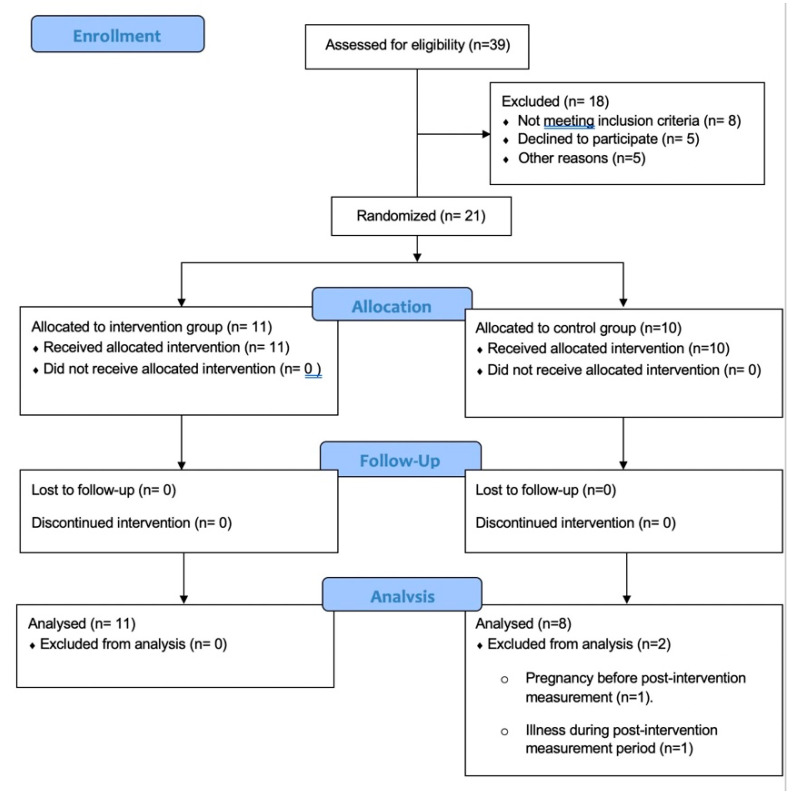
Shows the recruitment process.

**Figure 2 ijerph-18-12612-f002:**
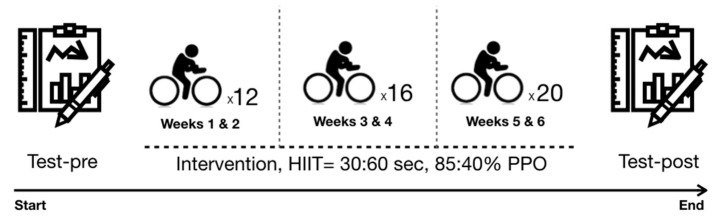
Represents the study timeline.

**Table 1 ijerph-18-12612-t001:** Participants.

	HIIT (*n* = 11)		Control (*n* = 8)	
	Pre	Post	Pre	Post
Age (years)	38 ± 5.5	-	35 ± 8.2	-
Sex (male/female)	6 m/5 f	-	4 m/4 f	-
Duration of T1DM (years)	20.58 ± 8.4	-	21.16 ± 6.5	-
BMI (kg/m^2^)	25.1 ± 0.4	24.9 ± 0.6	25.2 ± 0.8	25.3 ± 0.6
HbA1 (mmol/mol)	58.1 ± 15.3	54.9 ± 11.6	59.1 ± 10.1	59.3 ± 9.5
VO_2_max. (mL/min/kg)	37.1 ± 4.1	40.4 ± 3.8	37.0 ± 5.5	37.2 ± 5.1

**Table 2 ijerph-18-12612-t002:** Health-related quality of life.

	HIIT (*n* = 11)		Control (*n* = 8)		
	Pre	Post	Pre	Post	ES
Physical functioning (PF)	96.3 ± 3.2	98.1 ± 2.5 *	98.7 ± 2.3	99.3 ± 1.7	0.24
Physical role limitations (PR)	34.0 ± 12.6	61.3 ± 17.2 *	37.5 ± 13.3	31.2 ± 11.5 ^2^	0.66
Pain	44.5 ± 11.2	61.8 ± 7.5 *	37.5 ± 11.6	45.0 ± 14.1 ^2^	1.53
General health (GH)	49.5 ± 6.1	57.7 ± 5.1 *	51.25 ± 5.2	55.6 ± 6.2	0.81
Energy/fatigue (E)	23.1 ± 4.0	28.1 ± 4.6 *	23.7 ± 3.5	22.5 ± 4.6 ^2^	0.24
Social functioning (SF)	53.4 ± 5.8	71.6 ± 5.8 *	54.6 ± 9.3	60.9 ± 4.4 ^2^	0.87
Emotional role limitations (ER)	84.8 ± 17.4	96.9 ± 10.0	83.3 ± 17.8	87.5 ± 17.2	0.19
Emotional well-being (EWB)	57.8 ± 6.0	60.7 ± 3.5	58.0 ± 3.7	59.0 ± 2.8	0.24

Data are presented by mean ± standard deviation, * Significant difference between pre–post, ^2^ Significant difference between groups in that measurement point (pre or post), ES: effect size (Cohen’s d), *p* < 0.05.

**Table 3 ijerph-18-12612-t003:** Sleep quality.

	HIIT (*n* = 11)		Control (*n* = 8)		
	Pre	Post	Pre	Post	ES
Sleep Quality	6.1 ± 0.5	4.3 ± 0.5 *	5.5 ± 0.9	5.1 ± 0.6 ^2^	0.54

Data are presented by mean ± standard deviation, * Significant difference between pre–post, ^2^ Significant difference between groups in that measurement point (pre or post), ES: effect size (Cohen’s d), *p* < 0.05.

**Table 4 ijerph-18-12612-t004:** Exercise motivation.

	HIIT (*n* = 11)		Control (*n* = 8)		
	Pre	Post	Pre	Post	ES
Intrinsic	16.2 ± 1.2	18.3 ± 0.5 *	16.7 ± 1.5	17.6 ± 0.9 ^2^	0.96
Identified	15.9 ± 1.2	17.2 ± 0.9 *	16.3 ± 0.7	16.6 ± 0.5 ^2^	0.82
Introjected	8.1 ± 0.8	8.0 ± 0.8	7.6 ± 0.7	7.7 ± 0.7	0.39
External	10.1 ± 1.3	6.6 ± 0.9 *	10.1 ± 1.3	9.7 ± 1.4 ^2^	2.63
Amotivation	8.8 ± 0.7	4.7 ± 0.9 *	8.7 ± 0.4	8.2 ± 0.8 ^2^	4.1

Data are presented by mean ± standard deviation, * Significant difference between pre–post, ^2^ Significant difference between groups in that measurement point (pre or post), ES: effect size (Cohen’s d), *p* < 0.05.

**Table 5 ijerph-18-12612-t005:** Exercise enjoyment.

	HIIT (*n* = 11)		Control (*n* = 8)		
	Pre	Post	Pre	Post	ES
Enjoyment	72.5 ± 3.5	75.5 ± 3.4 *	72.2 ± 3.0	72.7 ± 2.5 ^2^	0.94

Data are presented by mean ± standard deviation, * Significant difference between pre–post, ^2^ Significant difference between groups in that measurement point (pre or post), ES: effect size (Cohen’s d), *p* < 0.05.

## Data Availability

Direct to authors.

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
