# Peer review of "Effect of High-Intensity Interval Training on Quality of Life, Sleep Quality, Exercise Motivation and Enjoyment in Sedentary People with Type 1 Diabetes Mellitus"

_ijerph, 2021, doi:10.3390/ijerph182312612_

Round 1
Reviewer 1 Report
This article proposes the HIIT as the potential exercise program which improves health-related quality of live, sleep quality, exercise motivation and enjoyment while preventing post-exercise hypoglycemia event. I have a few issues to be addressed.
- The authors described that the control group were asked to maintain their current lifestyle and dietary intakes during the study period. I wonder that some of them have their own exercise habit.
- I wonder why the authors did not try comparing between the HIIT and moderate intensity continuous training.
- An illustration about HIIT would be very helpful for readers to understand the whole scheme of training procedures.
Author Response
Reviewer 1
This article proposes the HIIT as the potential exercise program which improves health-related quality of live, sleep quality, exercise motivation and enjoyment while preventing post-exercise hypoglycemia event. I have a few issues to be addressed.
- The authors described that the control group were asked to maintain their current lifestyle and dietary intakes during the study period. I wonder that some of them have their own exercise habit.
Exercise habits were checked by the IPAQ to ensure the sedentary behavior of the participants.
“Furthermore, they completed two questionnaires before the beginning of the measurement protocols: the PAR-Q to assess participants’ level of risk to safely participate and the IPAQ (short version), to ensure the previous sedentary behavior of the subjects”. (Lines 95-98)
- I wonder why the authors did not try comparing between the HIIT and moderate intensity continuous training.
Moderate intensity continuous training was discarded because would require a complex insulin dosage adjustment that logistically had been very complicated to conduct correctly in all the participants of that group.
- An illustration about HIIT would be very helpful for readers to understand the whole scheme of training procedures.
Added
Reviewer 2 Report
The subject of the study is interesting and deserves consideration. However, the paper suffers from the lack of any diabetologists from the Authors' list, who could have added much to the methods (insulin dosage adjustment protocols, glucose checks, and meal adjustments at night, for instance). his is also a severe limitation of the study which, in case the paper is accepted for publication has to be mentioned appropriately
Author Response
Reviewer 2
The subject of the study is interesting and deserves consideration. However, the paper suffers from the lack of any diabetologists from the Authors' list, who could have added much to the methods (insulin dosage adjustment protocols, glucose checks, and meal adjustments at night, for instance). This is also a severe limitation of the study which, in case the paper is accepted for publication has to be mentioned appropriately.
This consideration is very valuable and completely reasonable. However, the fact that medical specialists were not included in the research group is due to the fact that the adjustments to be made were not going to be complex as detailed in the methods section. Despite this, due to the participants' illness, the University medical services were aware of the development of the research and of the research schedules to guarantee the safety of the participants.
“No medical specialist were included in the research group which would have contributed to a better quality in the metabolic adjusts. However, the medical services of the university were aware of the investigation development”(Lines 363-365)
Reviewer 3 Report
I have the following comments to make:
- The authors should better define their inclusion/ exclusion criteria. How did they define sedentary lifestyle? What about regular smoking? What are the comorbidities NOT related to diabetes that they would exclude T1DM patients from this study? What about the presence of diabetic complications on these patients? Were there obese participants or not? Were the patients under both injections and pump therapy? Were there any data regarding the compliance of these patients in the management of T1DM a priori? Although any effect of these parameters was expected to be overcome by random allocation of patients to each study group, these data still have to be provided.
- Have the authors measured data regarding the level of exercise/ dietetic habits of these patients prior to inclusion or during their participation in the study? Or they have just asked from their participant to maintain them at the same level as previously? The authors are suggested to confirm and clearly state that these data remained unchanged/ were not different between groups during the study period. If these data are not available, the authors should at least include this information as an additional limitation of their study.
- Regarding the 24-h monitoring of patients for hypoglycaemia, how frequently patients were asked to measure BG levels? Or did they only have to do that in case of feeling unwell? Is there any case that perception of hypoglycaemia was disturbed in anyone of the study participants?
- Did the author make sample size calculations to determine the appropriate number of patients needed to answer their research question?
- In their results, the authors should first provide some baseline data regarding their participants including age, sex, bmi, duration of diabetes, hba1c, smoking status, diabetic complications (if any), or even details from previous psychological assessment (if available) etc, preferably as a table. Please also compare these data between the two groups using statistical criteria. Is there any case these differences (if any) to have influenced the results?
- The authors are suggested to clarify what exactly their statistically significant differences refer to. In my opinion, they should not only make comparisons between pre- and post-intervention measurements separately for each group; they should also compare the between and within groups differences (group*intervention interaction). They should also compare the pre-intervention measurements between study groups to confirm that both groups are similar a priori. It could also be helpful for the average reader if authors explained what the interpretation of the ES value given is. Last but not least, is it correct to say that comparison between pre-and post-intervention measurements in each group was performed with both Mann Whitney test and Wilcoxon test? Isn’t the Wilcoxon test only the appropriate criterion?
- The authors should also clearly provide their results on the differences in hypoglycaemia frequency and severity between the two study groups. That’s because they comment on that finding in their discussion, however without having previously given these results to the readers.
- As the number of 70 references is great enough, especially on a field which is new, the authors are suggested to try to decrease the total number of references by selecting those references that are more recent, relevant and significant.
- Few grammar/syntax mistakes exist throughout the text. The authors are suggested to revise their manuscript ideally with the help of a native English speaker.
- Briefly the results of the present study have to be included in the abstract as well.
Author Response
Reviewer 3
I have the following comments to make:
- The authors should better define their inclusion/ exclusion criteria. How did they define sedentary lifestyle? What about regular smoking? What are the comorbidities NOT related to diabetes that they would exclude T1DM patients from this study? What about the presence of diabetic complications on these patients? Were there obese participants or not? Were the patients under both injections and pump therapy? Were there any data regarding the compliance of these patients in the management of T1DM a priori? Although any effect of these parameters was expected to be overcome by random allocation of patients to each study group, these data still have to be provided.
- Sedentary lifestyle was defined by IPAQ.
- This point refers to diseases that are not diabetes (respiratory, musculoskeletal, cardiovascular ...).
- This point was exposed in the manuscript: “Subjects excluded from the study include those who smoke regularly”.
- Point 5 of inclusion criteria aimed to include diabetes complications as well such a determinant factor. It should be showed properly in the manuscript. “no known comorbidities related or not to diabetes” (Line 89)
- Participants were not obese. Data of height and weight from the methods aimed to show this fact. It should be showed properly in the manuscript. “or suffer obesity” (Line 91)
- Due to the adjustments did not include insulin modifications, we believed that providing information about insulin treatment would be useless.
- Data about previous DT1 control were not asked. However, time since diabetes debut and absence of diabetes complications were required to ensure these facts.
- Have the authors measured data regarding the level of exercise/ dietetic habits of these patients prior to inclusion or during their participation in the study? Or they have just asked from their participant to maintain them at the same level as previously? The authors are suggested to confirm and clearly state that these data remained unchanged/ were not different between groups during the study period. If these data are not available, the authors should at least include this information as an additional limitation of their study.
Level of exercise was measured by IPAQ and participants were asked to maintain the same habits of physical activity. Regarding diet, data were not extracted due to the complexity of measurement and participants were asked to keep their nutritional habits with no change. This point will be included in limitations section: “Nutritional habits of the participants were not measured despite were asked to maintain it with no change”. (Lines 370-371)
- Regarding the 24-h monitoring of patients for hypoglycaemia, how frequently patients were asked to measure BG levels? Or did they only have to do that in case of feeling unwell? Is there any case that perception of hypoglycaemia was disturbed in anyone of the study participants?
Due to all the participants had systems of blood glucose measurement with no puncture, they could check their glucose level of any previous hour with no problems, despite feeling well.
- Did the author make sample size calculations to determine the appropriate number of patients needed to answer their research question?
In an a priori analysis of the required sample size (G*Power V.3.1.9.6), we needed 12 subjects per group, we had 11 participants in the experimental group and eight in the control group because it was impossible to recruit more participants with the desirable characteristics.
However, a statistical power analysis performed with G*power 3.0 resulted in values above 0.80 for all variables and tests performed, so we can conclude that the results are robust. Nevertheless, we have added this fact as a limitation of the study in the corresponding section.
“Although the statistical power analysis performed with G*power 3.0 resulted in values higher than 0.80 for all variables and tests performed, the small sample size remains the main limitation” (Lines 365-367)
- In their results, the authors should first provide some baseline data regarding their participants including age, sex, bmi, duration of diabetes, hba1c, smoking status, diabetic complications (if any), or even details from previous psychological assessment (if available) etc, preferably as a table. Please also compare these data between the two groups using statistical criteria. Is there any case these differences (if any) to have influenced the results?
|
|
HIIT (N=11) |
CONTROL (N=8) |
||
|
|
Pre |
Post |
Pre |
Post |
|
Age (years) |
38±5.5 |
- |
35±8.2 |
- |
|
Sex (male/female) |
6 m/5 f |
- |
4 m/ 4 f |
- |
|
Duration of T1DM (years) |
20.58±8.4 |
- |
21.16±6.5 |
- |
|
BMI (kg/m2) |
25,1 ±0,4 |
24,9±0,6 |
25,2 ±0,8 |
25,3±0,6 |
|
HbA1 (mmol/mol) |
58.1 ± 15.3 |
54.9 ± 11.6 |
59.1 ± 10.1 |
59.3± 9.5 |
|
VO2max. (ml/min/kg) |
37.1 ± 4.1 |
40.4 ± 3.8 |
37.0 ± 5.5 |
37.2 ± 5.1 |
We consider important to explain that some of the variables mentioned belong to other studies already published from this research and that in this study we aimed to point psychological variables despite this information is very important in this study as well.
- Alarcón-Gómez, J., Calatayud, J., Chulvi-Medrano, I., & Martín-Rivera, F. (2021). Effects of a HIIT Protocol on Cardiovascular Risk Factors in a Type 1 Diabetes Mellitus Population. International journal of environmental research and public health, 18(3), 1262. https://doi.org/10.3390/ijerph18031262
- Alarcón-Gómez, J., Martin Rivera, F., Madera, J., & Chulvi-Medrano, I. (2020). Effect of a HIIT protocol on the lower limb muscle power, ankle dorsiflexion and dynamic balance in a sedentary type 1 diabetes mellitus population: a pilot study. PeerJ, 8, e10510.https://doi.org/10.7717/peerj.10510
- The authors are suggested to clarify what exactly their statistically significant differences refer to. In my opinion, they should not only make comparisons between pre- and post-intervention measurements separately for each group; they should also compare the between and within groups differences (group*intervention interaction). They should also compare the pre-intervention measurements between study groups to confirm that both groups are similar a priori.
It could also be helpful for the average reader if authors explained what the interpretation of the ES value given is. Last but not least, is it correct to say that comparison between pre-and post-intervention measurements in each group was performed with both Mann Whitney test and Wilcoxon test? Isn’t the Wilcoxon test only the appropriate criterion?
The statistical analyzes mentioned have been developed but were not included in the manuscript. They have been included in all the variables of study. It shows that there were not differences a prior between groups. In the same way both Mann Whitney and Wilcoxon were conducted to check intra and inter-group differences in the variables which normality could not be assumed. Results of these analysis have been included as well.
- The authors should also clearly provide their results on the differences in hypoglycaemia frequency and severity between the two study groups. That’s because they comment on that finding in their discussion, however without having previously given these results to the readers.
The data has been added in results section.
3.5. Adverse events.
“There were three mild hypoglycemia cases (67.9 ± 2.6 mg/dl) of 198 total trainings (1.5%), occurring immediately after exercise which only required a few minutes of rest and carbohydrate ingestion to be solved. No adverse cardiac events, respiratory events or musculoskeletal injuries were reported in the experimental period. There were no episodes of hyperglycemia, nocturnal hypoglycemia or episodes of diabetic ketoacidosis” (Lines 268-272)
- As the number of 70 references is great enough, especially on a field which is new, the authors are suggested to try to decrease the total number of references by selecting those references that are more recent, relevant and significant.
Bibliography has been filtered to eliminate redundant or non-significant studies.
- Few grammar/syntax mistakes exist throughout the text. The authors are suggested to revise their manuscript ideally with the help of a native English speaker.
A native English speaker have analyzed the manuscript and few grammar/syntax mistakes have been solved.
- Briefly the results of the present study have to be included in the abstract as well.
Results have been added in the Abstract section.